# Lateral carbon export has low impact on the net ecosystem carbon balance of a polygonal tundra catchment

Lutz Beckebanze[1,2], Benjamin R. K. Runkle[1,3], Josefine Walz[1,4], Christian Wille[5], David Holl[1,2], Manuel Helbig[1,6], Julia Boike[7,8], Torsten Sachs[5], and Lars Kutzbach[1,2]

[1]Institute of Soil Science, Universität Hamburg, Germany
[2]Center for Earth System Research and Sustainability (CEN), Universität Hamburg, Germany
[3]Department of Biological and Agricultural Engineering, University of Arkansas, Fayetteville, AR, USA
[4]Climate Impacts Research Centre, Institute for Ecology and Environmental Science, Umeå University, 98107 Abisko
[5]Helmholtz-Zentrum Potsdam – Deutsches Geo Forschungs Zentrum (GFZ), Potsdam, Germany
[6]Department of Physics & Atmospheric Science, Dalhousie University, Halifax, Canada
[7]Alfred Wegener Institute Helmholtz Centre for Polar and Marine Research, Potsdam, Germany
[8]Department of Geography, Humboldt-Universität zu Berlin, Germany

**Correspondence:** Lutz Beckebanze (lutz.beckebanze@uni-hamburg.de)

**Abstract.** Permafrost-affected soils contain large quantities of soil organic carbon (SOC). Changes in the SOC pool of a particular ecosystem can be related to its net ecosystem carbon balance (NECB) in which the balance of carbon (C) influxes and effluxes is expressed. For polygonal tundra landscapes, accounts of ecosystem carbon balances in the literature are often solely based on estimates of vertical carbon fluxes. To fill this gap, we present data regarding the lateral export rates of dissolved
inorganic carbon (DIC) and dissolved organic carbon (DOC) from a polygonal tundra site in the North-Siberian Lena River Delta, Russia. We use water discharge observations in combination with concentration measurements of water-borne carbon to derive the lateral carbon fluxes from one growing season (2 June–8 September 2014 for DOC, 8 June–8 September 2014 for DIC). To put the lateral C fluxes into context, we furthermore present the surface–atmosphere eddy covariance fluxes of carbon dioxide ($CO_2$) and methane ($CH_4$) from this study site.

The results show cumulative lateral DIC and DOC fluxes of 0.31–0.38 g m$^{-2}$ and 0.06–0.08 g m$^{-2}$, respectively during the 93-day observation period (8 June–8 September 2014). Vertical turbulent fluxes of $CO_2$-C and $CH_4$-C accumulated to -19.0 $\pm$ 1.2 g m$^{-2}$ and 1.0 $\pm$ 0.02 g m$^{-2}$ in the same period. Thus, the lateral C export represented about 2% of the net ecosystem exchange of $CO_2$ (NEE). However, the relationship between lateral and surface–atmosphere fluxes changed over the observation period. At the beginning of the growing season (early June), the lateral C flux outpaced the surface-directed
net vertical turbulent $CO_2$ flux, causing the polygonal tundra landscape to be a net carbon source during this time of the year. Later in the growing season, the vertical turbulent $CO_2$ flux dominated the NECB.

**Keywords.** net ecosystem carbon balance, dissolved inorganic carbon, dissolved organic carbon, polygonal tundra

# 1 Introduction

Permafrost regions have accumulated $1,300 \pm 200$ Pg of soil organic carbon (SOC), of which $472 \pm 27$ Pg are stored within the top 1 m of soil (Hugelius et al., 2014). In a warming climate, previously frozen SOC can be mobilised and lost from the permafrost-affected ecosystems through vertical and lateral carbon fluxes. Many studies (e.g., Koven et al., 2015; Schuur et al., 2015) focus on vertical gaseous carbon (C) fluxes in the form of the greenhouse gases (GHGs) carbon dioxide ($CO_2$) and methane ($CH_4$). However, C loss can also occur laterally as dissolved organic carbon (DOC) and dissolved inorganic carbon (DIC), which is exported through runoff of meltwater and rainwater (e.g., Fouché et al., 2017; Olefeldt and Roulet, 2014) and may be emitted in the form of GHGs to the atmosphere outside of the spatial observation range (such as in coastal regions as shown by Lougheed et al., 2020). Thus, to estimate the net ecosystem carbon balance (NECB), both vertical and lateral C fluxes must be considered (Chapin et al., 2006). Although scholars have identified lateral C transport as an important mechanism of C losses from terrestrial ecosystems in the Arctic (Zhang et al., 2017), little is known about the contribution of lateral C fluxes to the NECB. So far, lateral C fluxes have only been included in NECB estimations in two subarctic catchments in Northern Sweden. In the first catchment, lateral C fluxes contribute 6–15% (Chi et al., 2020) and 4–28% to the annual NECB (Öquist et al., 2014) and in the second catchment, lateral C fluxes represent 35% of the NECB (Lundin et al., 2016). To our knowledge, there has been no attempt yet to combine the lateral and vertical C fluxes in an Arctic ecosystem.

Here, we estimate the NECB for a Siberian Arctic tundra ecosystem and present the individual flux contributions during one growing season. Since 2002, the vertical C fluxes of $CO_2$ and $CH_4$ have been observed at our study site (Holl et al., 2019) using the eddy covariance method (Baldocchi, 2003). The study site, located in the North-Siberian Lena River Delta, is characterized by polygonal lowland tundra landscape. In this study, we combine the vertical C fluxes ($F_{CO_2}$ and $F_{CH_4}$) with the lateral C fluxes ($F_{DOC}$ and $F_{DIC}$) to derive the NECB for one growing season in 2014. We also compare the temporal dynamics of DIC and DOC concentrations with respect to the water discharge rate to find a potential driver for the concentrations of dissolved carbon. In summary, this study examines two research questions: (1) What is the influence of the lateral waterborne C fluxes on the NECB of the polygonal tundra landscape? (2) How do DOC and DIC concentrations and fluxes develop over the growing season?

# 2 Methods

## 2.1 Study site

The study site, Samoylov Island (Fig. 1), which is located in the southern part of the Russian Lena River Delta, consists of two geomorphological units: a modern floodplain in the west ($\sim 1.5$ km$^2$) and a late-Holocene river terrace in the east ($\sim 3$ km$^2$, Boike et al., 2013). The floodplain and river terrace are at elevations of 0–8 m and 8–13 m meters, respectively, above the water level of the Lena River (Boike et al., 2012a). Rain and meltwater from the river terrace drain towards the Lena River and the lower-lying floodplain. Polygonal tundra with a shallow active layer (less than 1 m) and wet to moist tundra vegetation (sedges, mosses, dwarf shrubs) characterizes the river terrace. Following the World Reference Base for Soil Resources (WRB,

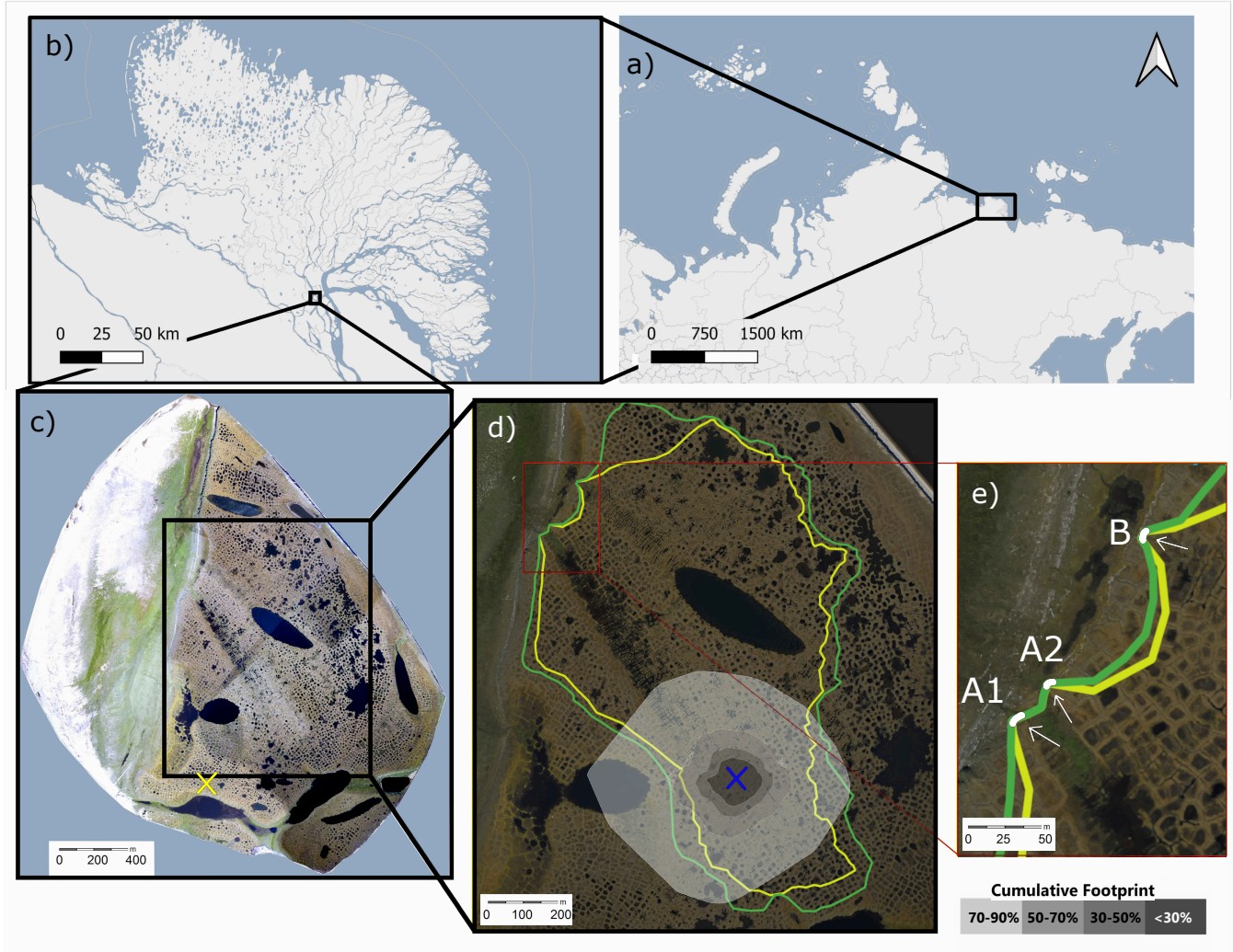

**Figure 1.** Top: Map of Northern Russia (a) and the Lena River Delta (b). Bottom: Map of Samoylov Island (c) with the maximum (green) and minimum (yellow) catchment size (d). The blue cross in (d) marks the eddy covariance tower's location and the cumulative footprint is shown in gray shades. 30 % of the flux likely originated from within the dark gray area, 50 % from within the medium-dark gray area, 70 % from within the medium-light gray area and 90 % from within the light gray area. The yellow cross in (c) denotes the location of the meteorological station with the active layer depth observations. Panel (e) shows the locations of outflows $A_1$, $A_2$, and B. Map data from © OpenStreetMap contributors 2020, distributed under the Open Data Commons Open Database License (ODbL) v1.0 (a and b) and modified after Boike et al. (2012b; c, d, and e).

2014), the main soil types include Histic Cryosols in polygon centers and Turbic Glacic Cryosols at elevated polygon rims on the river terrace (Pfeiffer and Grigoriev, 2002; Boike et al., 2013; Zubrzycki et al., 2014).

## 2.2 Catchment characteristics

We estimated the catchment size by analyzing a digital elevation model (DEM) of Samoylov Island (Boike et al., 2012a) in ArcMap 10.1 (Environmental Systems Research Institute , ESRI). This DEM has a vertical and horizontal accuracy of less than 1 m. Due to the low elevation gradient in some catchment areas, we also used field observations from 2019 and shoreline measurements during different stages of the spring flood in 2014 to validate our estimations of the catchment size. To distinguish flow paths, we furthermore used the orthomosaic of Samoylov Island produced by Boike et al. (2012b) with an average horizontal resolution of 0.33 m. Based on this methodology, we estimated a larger catchment size than Helbig et al. (2013). This catchment drains through three outflows ($A_1$, $A_2$, and B). We estimated a catchment size of 0.69—0.84 km$^2$ (Fig. 1, panel (d); green denotes the maximum estimate, while yellow denotes minimum estimate).

The polygonal tundra in this catchment is characterized by intact and degraded ice-wedge polygons with water-filled centers as well as polygons with dry centers. Water-filled troughs between the polygons are also present (Kartoziia, 2019). A low-lying and and largely inundated area stretches from the center of the catchment towards the outflows $A_1$ and $A_2$ (Fig 1, panel (d)).

## 2.3 Water discharge

Between 2 June and 8 September 2014, we measured the water level at outflows $A_1$, $A_2$, and B using pressure sensors (Mini-Diver, Schlumberger Water Services, Netherlands) placed within 10 cm of the weir wall outside of the zone of fast flowing water. To calculate the water level above the diver, we subtracted the barometric pressure from the diver pressure. We measured the barometric pressure at the eddy covariance tower (CS100, Campbell Scientific, USA). Prior to the water discharge rate calculation, we corrected the water level estimations with a linear relationship with manual water level measurements (obtained using a ruler). The water discharge rates at outflows $A_1$ and $A_2$ were observed with V-notch weirs as described by Helbig et al. (2013). To calculate the water discharge rates, we used Kulin and Compton's 1975 method for calculating V-notch weirs: $Q_a = \frac{8}{15} C_{weir} \sqrt{2g}\, tan\left(\frac{1}{2}\alpha\right) h^{\frac{3}{2}}$, where $Q_a$ is the discharge rate in Ls$^{-1}$, $h$ is the water level above the notch determined from the pressure sensor in feet (ft), $C_{weir} = 0.58$ is the dimensionless weir constant, $g = 9.81$ ms$^{-2}$ is the gravitational acceleration, and $\alpha = 60°$ is the angle of the V-notch. After performing the water discharge rate calculations, we validated the estimated water discharge rates with manual bucket measurements (with a stop watch and a defined bucket volume).

Outflow B was located about 150 meters north of the A-outflows and the discharge rate was measured using a long-throated flume (RBC flume, 13.17.02, Eijkelkamp Agrisearch Equipment, the Netherlands). We calculated $Q_b$ using the manufacturer's equation for this RBC flume: $Q_b = -0.066 + 0.016\, h + 0.00063\, h^2 + 7 \cdot 10^{-7}\, h^3$, where $h$ is the water level above the notch in mm and $Q_b$ is the water discharge rate in Ls$^{-1}$.

## 2.4 Dissolved inorganic carbon (DIC)

Between 8 June and 8 September 2014, we deployed a $CO_2$ sensor in the water column at outflow $A_1$ ($CO_2$ measurement system with multisensor module MSM-S2, UIT GmbH, Germany). This sensor measured the concentration of dissolved $CO_2$ ($C_{dCO_2}$) every 5 minutes (each measurement 15 seconds long).

DIC consists of dissolved $CO_2$ (as free $CO_2$ and carbonic acid; $[H_2CO_3]$), bicarbonate ions ($HCO_3^-$), and carbonate ions ($CO_3^{-2}$). In a freshwater system, each component's contribution to the DIC concentration depends on water temperature and pH; the bicarbonate equilibrium describes this relationship. More details on the bicarbonate equilibrium can be found in the book by Dodds and Whiles (2010). We calculated the carbonic acid concentration ($C_{HCO_3^-}$) from $C_{dCO_2}$, water temperature, and pH: $C_{HCO_3^-} = \frac{K_1 \cdot C_{dCO_2}}{a(H^+)}$ where $a(H^+) = 10^{-pH}$. The dimensionless value of $K_1$ is temperature-dependent, and following Wong and Hsu (1991), is described as $K_1 = 10^{15.11 - 0.034 \cdot T - 3406.12 \cdot T^{-1}}$, where $T$ describes the water temperature in the unit $K$.

The pH value was frequently, but not continuously measured throughout the 2014 growing season (N=40). To fill the gaps in the pH time series, we applied a running mean. The pH values varied between 6.60 and 6.99; therefore, the contribution of carbonate to $C_{DIC}$ was not relevant. Due to the negligible amount of carbonate, we calculated $C_{DIC}$ as the sum of $C_{HCO_3^-}$ and $C_{dCO_2}$.

The $C_{dCO_2}$ sensor failed to record accurate measurements between 18 July and 30 July, and thus we excluded the recorded values during this period. To fill this data gap, we applied an artificial neural network (ANN), targeting $C_{dCO_2}$ and using four input parameters (air temperature, relative air humidity, vertical $CO_2$ flux, and DOC concentration). We set up the ANN as a multilayer perceptron with 10 hidden neurons in Matlab's Deep Learning Toolbox (MATLAB, 2019b) using Levenberg-Marquardt backpropagation as an optimization algorithm. We divided the data sets into training (70%), validation (15%), and testing (15%) subsets.

## 2.5 Dissolved organic carbon (DOC)

We routinely analyzed unfiltered water samples (n = 126) from all three outflows using a portable UV–Vis spectrometer probe (spectro::lyzer, s::can Messtechnik GmbH, Austria). The measurements were supported by lab analyses to calibrate the spectrometer probe observations and increase data availability (n = 41). Water samples for calibration were collected in acid-washed glass bottles, acidified to a pH value of 2, cooled to 4 °C for transport, and filtered (40 $\mu$m) prior to analysis. Analysis was conducted using a total organic carbon (TOC) analyzer (TOC-L, Shimadzu, Japan). We estimated $C_{DOC}$ from unfiltered water samples following the workflow presented by Avagyan et al. (2014). This approach is based on the finding that different absorbance bands of a spectrometer probe can be suitable for the description of the DOC concentration, depending on the types of organic compounds in the sample water. We found a good agreement of $R^2_{adj} = 0.82$ between $C_{DOC}$ from the spectrometer probe and $C_{DOC}$ from the TOC analyzer (Fig. A3). Details of the method can be found in supplement A1.

## 2.6 DOC and DIC flux

The catchment area-normalized lateral carbon fluxes of DOC ($F_{DOC}$) and DIC ($F_{DIC}$) are the product of water discharge rate $Q$ and $C_{DOC}$ and $C_{DIC}$, respectively, divided by the area of the catchment: $F_{DOC} = Q \cdot C_{DOC}/a$ and $F_{DIC} = Q \cdot C_{DIC}/a$, where $a$ describes either the minimum or maximum estimated catchment size.

## 2.7 Environmental conditions

Precipitation and air temperature were recorded throughout the study period at the meteorological station in the southern part of the island in one-hour intervals; Boike et al. (2019) published these measurements (data obtained from Boike et al. (2019)). The growing degree days ($GDD_{10}$) were calculated as the sum of all positive differences between the daily mean air temperature and the reference temperature (defined as 10 °C). The thaw depth was measured at a 150 grid point array next to the meteorological station by pushing a metal rod vertically into the ground (Boike et al., 2019) and was obtained from the GTNP database (GTNP Database, 2019).

## 2.8 Eddy covariance flux

We estimated the net vertical fluxes of $CO_2$ ($F_{CO_2}$), $CH_4$ ($F_{CH_4}$), and evapotranspiration (ET) using an eddy covariance (EC) measurement system. Holl et al. (2019) described raw data processing of $CO_2$ fluxes; and the gap-filled time series were obtained from Holl et al. (2018). High-frequency fluctuations in $CH_4$ concentration were observed with a Licor 7700 gas analyzer (Licor Biosciences, USA). Data processing of $CH_4$ fluxes followed the same method as described in Holl et al. (2019) for open-path $CO_2$ fluxes. Gap-filling of $CH_4$ fluxes was performed by applying a running mean of 48 hours. ET fluxes were observed using a Licor 7500A gas analyzer (Licor Biosciences, USA), and the data processing followed Helbig et al. (2013).

## 2.9 Cumulative fluxes

To quantify the impact carbon losses due to lateral transport have on the carbon balance of the catchment, we calculated the cumulative carbon fluxes of $CO_2$, $CH_4$, DIC, and DOC for the period between 8 June and 8 September 2014 in 30-minute intervals. Other flux components of the lateral C flux, e.g. particulate organic carbon or particulate inorganic carbon, are not accounted for in this study. Between 2 June and 7 June 2014, the component of $C_{DIC}$ was not obtained yet, therefore, this period is not included in the sums of carbon fluxes. However, this period is still part of the study period, since the spring flood had a great influence on the DOC flux dynamics.

## 2.10 Uncertainty estimation

In this study, uncertainties from random errors are indicated by the ± symbol, and the ranges of uncertainties from systematic errors are indicated with a hyphen (–).

We calibrated the observed water discharge rate $Q$ against and manual height measurements. Therefore, we assumed random errors from both pressure sensors to dominate the uncertainty of $Q$. According to the manuals, our used Diver pressure sensor has a typical accuracy of 0.05% at full scale (however, the error is not further specified). The atmospheric pressure sensor has an accuracy of ± 1 hPa (one standard deviation). We used the Gaussian error propagation to estimate the resulting uncertainty $u_Q$ following two steps. First, we estimated the resulting error of the height measurement $u_{p_h} = \sqrt{u_{p_d}^2 + u_{p_a}^2}$, where $u_{p_h}$ de-

scribes the uncertainty of the water level height measurement in hPa, and $u_{p_d}$ and $u_{p_a}$ describe the error of the Diver and the atmospheric pressure sensor, respectively. We converted $u_{p_h}$ to the unit of mm, $u_h$, by dividing through the density of water and the earth's gravitational force. Second, we estimated the resulting uncertainty of Q as $u_Q = \frac{\delta Q}{\delta h} \cdot u_h$, where $\frac{\delta Q}{\delta h}$ describes the partial derivative from Q with respect to h.

The uncertainty of the DOC concentration results from the limits of the TOC analyzer (TOC-L, Shimadzu, Japan). The manufacturer states a maximum error of 1.5% in repetitive measurements. We used the RMSE between the modeled DOC concentration from the spectrometer and the DOC concentration from the TOC analyzer to estimate DOC concentration's uncertainty, $u_{C_{DOC}}$.

For the estimation of the uncertainty of $C_{DIC}$ ($u_{DIC}$) we needed to consider the uncertainty of $C_{dCO_2}$, water temperature, and pH. According to the $C_{dCO_2}$ sensor's manual, the sensor has an accuracy of 5% and, after calibration, an offset of up to 1 mg L$^{-1}$. The accuracy of the water temperature probe is given as $u_{t_w} = 0.2$ K. We estimated the pH uncertainty from the standard deviation of multiple measurements of the same water sample. The overall uncertainty of $C_{DIC}$ was calculated using Gaussian error propagation as: $u_{DIC} = \sqrt{u_{dCO_2}^2 + u_{HCO_3}^2}$ with $u_{HCO_3} = \sqrt{(\frac{\delta C_{HCO_3}}{\delta C_{dCO_2}})^2 \cdot u_{dCO_2}^2 + (\frac{\delta C_{HCO_3}}{\delta pH})^2 \cdot u_{pH}^2 + (\frac{\delta C_{HCO_3}}{\delta T_w})^2 \cdot u_{T_w}^2}$.
We estimated the systematic and random uncertainty of the lateral C flux separately. Systematic uncertainty, described as $F_{DOC,sys}$ and $F_{DIC,sys}$, occurs due to systematic error of the catchment size and is estimated as $F_{DOC,sys} = F_{DOC_{a_{max}}} - F_{DOC_{a_{min}}}$, where $F_{DOC_{a_{min}}}$ and $F_{DOC_{a_{max}}}$ denote the DOC flux calculated with the largest and smallest assumed catchment size (resulting in the smallest and the largest DOC flux, respectively).

The random uncertainty of the lateral C flux, $F_{DOC,rand}$ and $F_{DIC,rand}$, resulting from random errors, is estimated as:

$$F_{DOC,rand} = F_{DOC} \sqrt{\left(\frac{u_Q}{Q}\right)^2 + \left(\frac{u_{C_{DOC}}}{C_{DOC}}\right)^2}.$$

We estimated the systematic uncertainty range of the cumulative lateral C flux ($\sum F_{DOC,sys}$ and $\sum F_{DIC,sys}$) as the difference between the cumulative fluxes with the smallest and the largest assumed catchment size:

$\sum F_{DOC,sys} = \Delta t \sum_{t_1}^{t_n} F_{i_{a_{max}}} - \Delta t \sum_{t_1}^{t_n} F_{i_{a_{min}}}$, where $\Delta t$ describes the duration of the measurement interval and $t_1$ and $t_n$ denote the first and the last time step of the measurement, respectively.

We estimated the random uncertainty of the cumulative lateral C flux ($\sum F_{DOC,rand}$ and $\sum F_{DIC,rand}$) as:

$$\sum F_{DOC,rand} = \Delta t \sqrt{\sum_{t_1}^{t_n} \left( F_{DOC} \sqrt{\left(\frac{u_Q}{Q}\right)^2 + \left(\frac{u_{C_{DOC}}}{C_{DOC}}\right)^2} \right)^2}.$$

For the uncertainty estimation of DIC, we replaced DOC with DIC in the four equations above. In instances in this text where only a range of lateral C flux is provided, we ignored the random uncertainty and focused on the dominant systematic uncertainty.

The uncertainty of the vertical EC fluxes $u_{F_{CO_2}}$ and $u_{F_{CH_4}}$ were estimated in the flux processing software *EddyPro* following Finkelstein and Sims (2001). Details on the flux uncertainty estimation of $F_{CO_2}$ can be found in Holl et al. (2019). We estimated the uncertainty of the cumulative vertical fluxes $u_{\sum F_{CO_2}}$ using the Gaussian error propagation for random uncertainties, resulting in $u_{\sum F_{CO_2}} = \Delta t \sqrt{\sum_{t_1}^{t_n} u_{F_{CO_2}}^2}$ and $u_{\sum F_{CH_4}} = \Delta t \sqrt{\sum_{t_1}^{t_n} u_{F_{CH_4}}^2}$ for $F_{CO_2}$ and $F_{CH_4}$, respectively.

## 3 Results

### 3.1 Environmental conditions

To put the observation year of 2014 into perspective, we compared the meteorological conditions at our study site between 8 June and 8 September 2014, to the meteorological conditions during the same 93-day period over 20 years (1998 – 2018, Fig. A1). With 87 °C at the end of the 93-day period, the growing degree days (GDD, Fig. A1 (a) in 2014 were among the average values, as established by the comparison dataset (67 $^{118}_{49}$ °C, Median $^{\text{75th Percentile}}_{\text{25th Percentile}}$). The 2014 thaw depths in the center and the rim of the polygons studied (Fig. A1 b and c) in 2014 were among the deepest recorded in a 17-year companion dataset (2002–2018).

The vertical water balance shows the precipitation, evapotranspiration, and water runoff rate between 2 June and 8 September 2014 (Fig. A2). The precipitation accumulation of 94 mm is within the average range for 14 of the available years between 1998–2018 (95 $^{138}_{76}$ mm, Median $^{\text{75th Percentile}}_{\text{25th Percentile}}$). In the same period, the evapotranspiration accumulated to 161 mm, and the lateral water runoff accumulated to 23–38 mm. The 2014 spring flood of the Lena River flooded parts of the catchment (field observation by B. Runkle). Therefore, the lateral water runoff was largely influenced by the out-flowing water of this spring flood, visible at the beginning of the observation period. The larger overall loss of water (183–189 mm) stands out more than the accumulation of water (94 mm) during the observation period. However, one component has been neglected in this water balance: the snow accumulation in winter, which was not observed between 2013 and 2014. However, in the winter of 2008 – 2009, there was a mean snow accumulation of 65±35 mm on Samoylov Island (snow water equivalent; Boike et al., 2013).

### 3.2 Lateral carbon flux dynamics

To focus more closely on lateral C flux dynamics, we examine the relationship between water discharge and DIC and DOC concentration (Fig. 2). At outflow $A_1$, high DIC concentrations were generally associated with low water discharge. With decreasing water discharge, the DIC concentration rose. A similar effect can be seen with the DOC concentration, which continuously increases as the water discharge rate decreases data recorded during the river flood are excluded. A comparison of the DIC and DOC concentrations shows that DIC concentrations were 4.31 $^{6.41}_{3.28}$ times higher than DOC concentrations (Median $^{\text{75th Percentile}}_{\text{25th Percentile}}$).

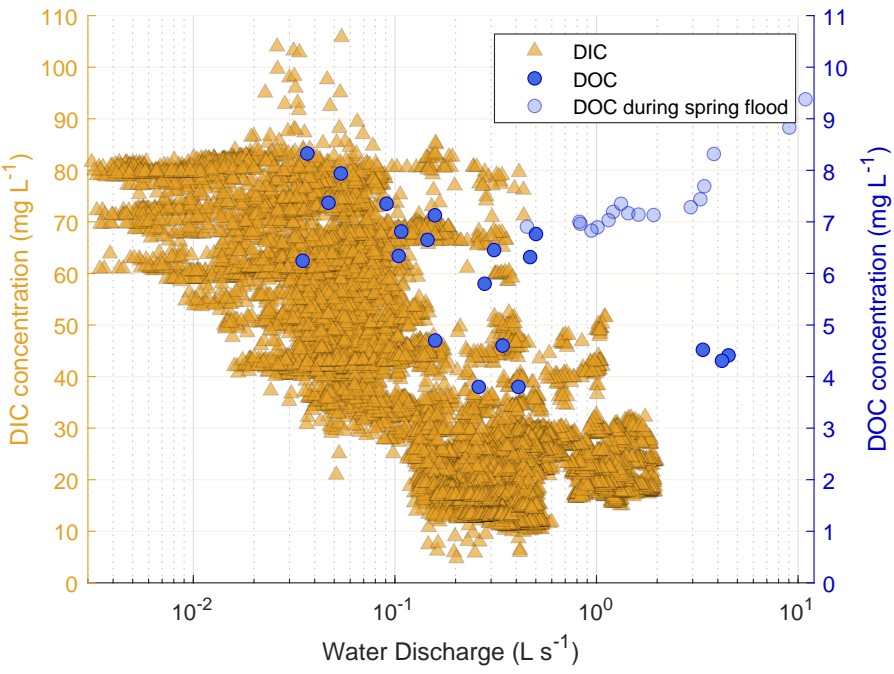

**Figure 2.** Dissolved inorganic carbon (DIC, triangles) and dissolved organic carbon (DOC, circles) concentrations (note the different scales) against the water discharge rate on a semi-logarithmic scale. DOC concentration is shown on the right y-axis, while data after the spring flood (June 2–8) are shown in transparent circles. The DIC concentration was not observed during the spring flood.

### 3.3 Net ecosystem carbon balance

In this section, we present the NECB for the study period consisting of the lateral ($F_{DOC}$ and $F_{DIC}$) and vertical carbon fluxes ($F_{CO2}$ and $F_{CH4}$). The cumulative fluxes of all NECB components between 8 June and 8 September are summarized in Fig. 3 and Tab. A1. Values with dominant systematic errors are expressed as ranges with an en-dash symbol, and values with dominant random errors are expressed with a plus-minus symbol.

During the 93-day period in 2014, the NECB accumulated to -17.6 – -17.5 ($\pm$ 1.2) g m$^{-2}$. The vertical fluxes of $F_{CO_2}$ and $F_{CH_4}$ contributed -19.0 $\pm$ 1.2 and 1.0 $\pm$ 0.02 g m$^{-2}$ to the NECB, respectively, while the lateral fluxes of $F_{DIC}$ and $F_{DOC}$ contributed 0.31 – 0.38 and 0.06 – 0.08 g m$^{-2}$ to the NECB, respectively, to the NECB. Thus, within the study period, lateral C fluxes exported 1.95 – 2.42% of the net ecosystem exchange (lateral-C-flux/NEE), i.e. the net C uptake due to the balance of photosynthesis and respiration.

We also split these cumulative fluxes into mean weekly fluxes (Fig. 4, (a) and (b), and Tab. A1 ). During the periodic spring flood that occurred in 2014 partially in the first week of June (2–7 June Fig. 4 (a), high lateral DOC flux (13.0 – 15.8 mg m$^{-2}$ d$^{-1}$) and CH$_4$-C flux (3.6 $\pm$ 0.3 mg m$^{-2}$ d$^{-1}$) outpaced the CO$_2$-C uptake (-7.0 $\pm$ 21.1 mg m$^{-2}$ d$^{-1}$) and indicate an ecosystem carbon source (positive NECB). During this period, $F_{DIC}$ was not yet observed. Therefore, the NECB is expected

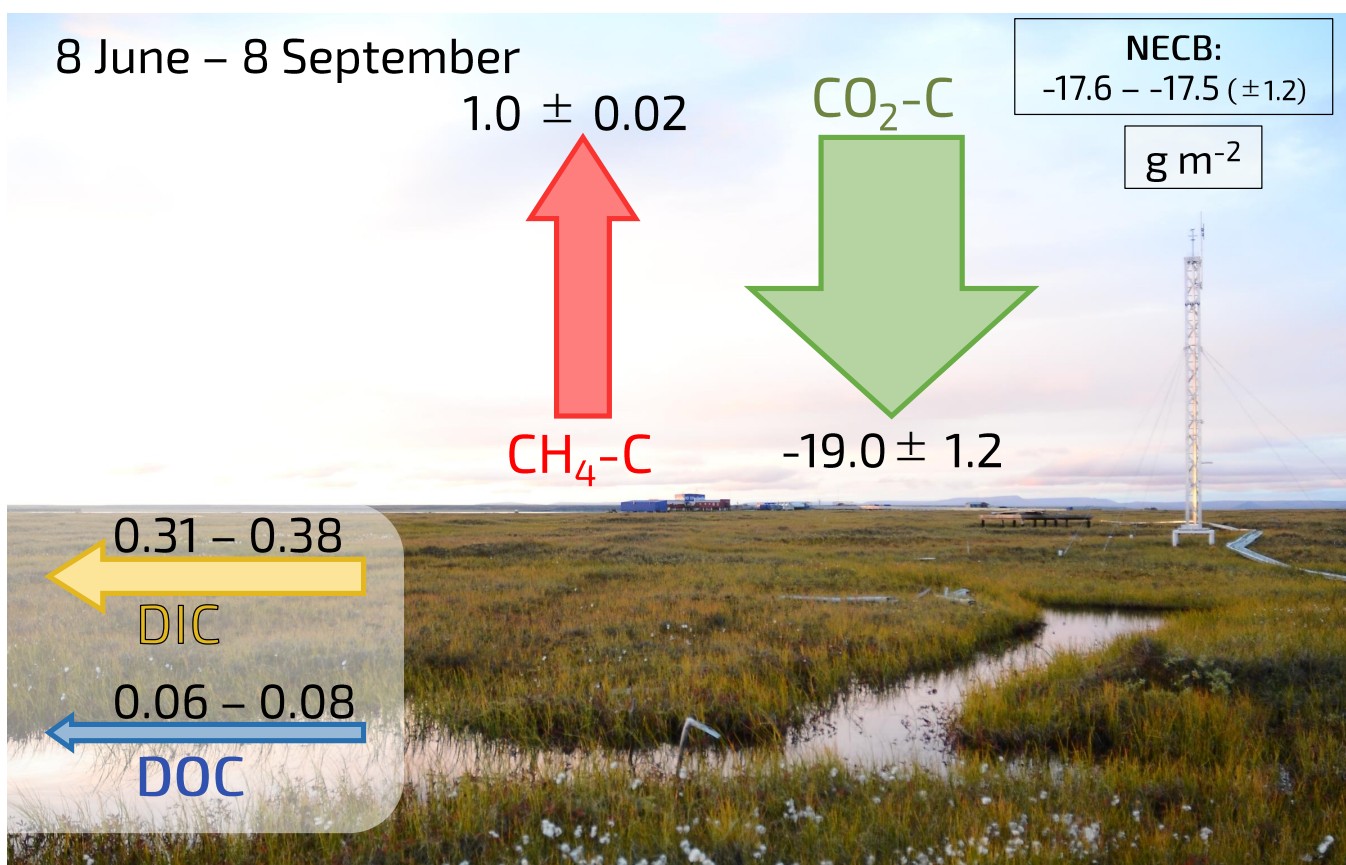

**Figure 3.** Schematic illustration of all four cumulative flux components of the NECB (DIC, DOC, $CH_4$-C, and $CO_2$-C) in g m$^{-2}$ during the study period in 2014. The NECB is shown in the top right corner. Uncertainties from systematic errors are denoted with an en dash (–), while uncertainties from random errors are indicated with the plus-minus symbol ($\pm$). The picture in the background was taken on 30 August 2016 and provided by Jean-Louis Bonne.

to be a stronger C source than presented in Fig. 4 (b).

From mid June until the beginning of August, the negative NECB indicates that the ecosystem served as a carbon sink due to high levels of plant $CO_2$ uptake. In August, the $CO_2$ sink strength decreased (Fig. 4) and the mean daily $CO_2$-C flux turned from negative to positive. At the same time, vertical $CH_4$ fluxes reached their maximum. Lateral DOC and DIC fluxes declined and the ecosystem acted as a weak carbon sink with an NECB also turned from negative to positive. During the eight September days within the study period, all fluxes acted as carbon sources. With a relative contribution of 97%, the $CO_2$-C emission dominated the NECB in September.

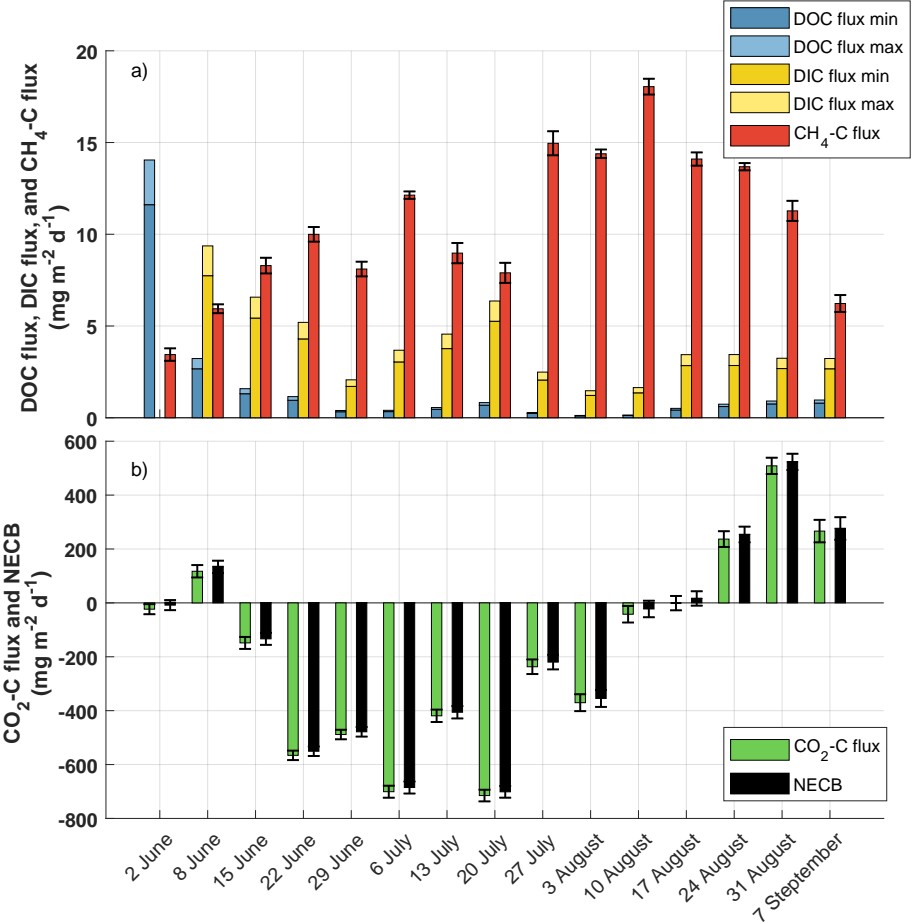

**Figure 4.** Seasonal development of NECB components as mean daily fluxes from 2 June (week 1) to 8 September (week 15). (a) Minor Components of NECB: DOC flux (blue), DIC flux (yellow), and $CH_4$-C flux (red); (b) Major component of NECB: $CO_2$-C flux (green) and the resulting NECB (black). Please note the different values of the y-axis in (a) and (b). Uncertainties from systematic errors are denoted with a second bar in a brighter color, while uncertainties from random errors are indicated with an error bar. The systematic uncertainty of the NECB is shown but not visible due to its small value.

## 4 Discussion

### 4.1 Comparison of DOC and DIC dynamics

We found a negative correlation between the water discharge rate and the DIC concentration (Fig. 2), meaning that higher water discharge rates dilute and decrease the DIC concentration. This result indicates that in other years, the DIC flux would not increase linearly with greater precipitation in other years. The results by Öquist et al. (2014) show precipitation to be an important driver of interannual variability in lateral C flux dynamics. Our study period had comparatively normal summer precipitation rates. In other years with higher precipitation rates, we would expect to find a higher water discharge rate. However,

based on our results, we do not expect a sharp rise in DIC flux to result from a higher water discharge rate. In one study, which focused on a catchment in Northern Sweden, a tripling in water discharge rate increased the annual lateral $^{14}$C-export by only 2% (Campeau et al., 2017). In contrast, in another study in Northern Sweden, annual DIC export increased exponentially with rising water discharge rates (Öquist et al., 2014). Similar to the DIC concentration, we also found a negative correlation between the water discharge rate and the DOC concentration when neglecting the period during the spring flood. This finding

suggests that higher discharge rates dilute and decrease the DOC concentration. Therefore, in seasons with higher discharge rates, the DOC flux would not rise linearly and the contribution of DOC export to the NECB probably would not rise. A similar correlation between DOC concentration and the water discharge rate has been reported in a palsa a subarctic catchment (Olefeldt and Roulet, 2012). However, DOC export from polygonal tundra may increase if arctic climate change will lead to accelerated degradation of ice-wedges, which is expected to enhance drainage of the permafrost landscape (Liljedahl et al.,

2016).

One unexpected finding was the relationship between DOC and DIC concentrations with a $C_{DIC}/C_{DOC}$ ratio of $4.31\,{}^{6.41}_{3.28}$ (Median $^{75\text{th Percentile}}_{25\text{th Percentile}}$). This ratio differs from those in other studies, which report a mean ratio of 0.65 from an Alaskan permafrost-affected watershed (Kling et al., 2000), a ratio of 0.24–1.30 in Canadian boreal biomes (Hutchins et al., 2019), and

250 a ratio of 0.28 in a mixed coniferous forest in Northern Sweden (Chi et al., 2020). However, one study reported a $C_{DIC}/C_{DOC}$ ratio of up to 11.6 in an ice-rich permafrost catchment in Northern Alaska (O'Donnell et al., 2019). A previous study at our study site found $C_{DIC}/C_{DOC}$ ratios between 6.6 and 15.5 at the island's northern floodplain outlet in September 2008 (Abnizova et al., 2012), which are higher values than we present in this study, with a ratio of $4.12\,{}^{4.42}_{3.82}$ in September 2014 (Median $^{75\text{th Percentile}}_{25\text{th Percentile}}$, not shown in the results). The high $C_{DIC}/C_{DOC}$ ratio hints to effective degradation and mineralization of

255 dissolved organic matter (DOM) in the surface waters of the studied catchment. Such effective degradation of DOM by either photo-oxidation (Cory et al., 2014, 2015) and/or microbial decomposition (e.g., Drake et al., 2015; Mann et al., 2015; Spencer et al., 2015) has been found and intensively studied in other arctic catchments. The studied polygonal tundra catchment is characterised by (1) a low relief and (2) mostly shallow water bodies (depth < 1 m). Both factors enhance decomposition and mineralization of DOM: The low relief leads to long residence times of DOM before export, and the shallowness of the water

bodies allows for intense light exposure and photodegradation of DOM, which, in turn, may promote microbial mineralization (Cory et al., 2015).

## 4.2 Net ecosystem carbon balance

We estimated the NECB using the lateral C fluxes (DIC and DOC flux) and the vertical fluxes of $CO_2$ and $CH_4$. Our results indicate that vertical $CO_2$ uptake dominated the NECB during the study period. The lateral C fluxes exported only 1.95 – 2.42% of the NEE. During the complete study period, we found the lateral C fluxes' contribution to the NECB to be smaller than the uncertainty range of the observed $CO_2$ uptake. Nevertheless, our results also show that lateral carbon loss can exceed vertical carbon uptake at the beginning of the growing season. This finding shows that lateral C fluxes can play an essential role in the NECB during intensive water runoff periods, as we show in Fig. 4 (a).

The question remains whether the resulting relationship between lateral C fluxes and the NECB would be similar in the previous and following growing seasons. A previous study at the site of the instant study includes a 15-year record of eddy covariance $CO_2$ fluxes between 19 July and August 23 of each year (Holl et al., 2019). It shows that $CO_2$ uptake in 2014 was among the lowest values in the 15-year record. In 12 other years, $CO_2$ uptake was stronger compared to the 2014 period. According to these data, we assume that the influence of lateral C fluxes on the NECB would have played an even less important role in many other years compared to 2014. In two studies from this site, researchers reported low but varying average $CH_4$-C fluxes in two summer seasons: in the first study, the $CH_4$-C fluxes vary between 7.5 mg m$^{-2}$ d$^{-1}$ (28 June–22 July 2004) and 17.3 mg m$^{-2}$ d$^{-1}$ (18 July–25 July 2003; Wille et al., 2008). In the second study, Beckebanze et al. (2022a) reported $CH_4$-C fluxes of $12.55\,^{16.07}_{9.65}$ mg m$^{-2}$ d$^{-1}$ (Median $^{\text{75th Percentile}}_{\text{25th Percentile}}$, 11 July–10 September 2019) Thus, our $CH_4$-C flux estimation of 10.6 mg m$^{-2}$ d$^{-1}$ in July 2014 lies within the range of estimates for other years. A study on DOC flux from a nearby island in the Lena River Delta reports a mean daily flux of 1.2 mg m$^{-2}$ d$^{-1}$ in July and August 2013 (Stolpmann et al., 2022). This estimate is higher than our estimations of 0.42–0.51 mg m$^{-2}$ d$^{-1}$ in July 2014, however, in the same order of magnitude compared to our estimates form June 2014 (1.51–1.83 mg m$^{-2}$ d$^{-1}$).

We also investigate the question of whether the measured EC flux would be representative of the entire catchment. Instruments at the EC tower were mounted at a height of 4.15 m, and the tower was located approximately 850 m southeast of the A-outflows (see Fig. 1). The normalized mean contributions of four surface classes (based on the classification by Muster et al., 2012) within the footprint of the EC flux amounted to 66% (dry tundra), 18% (wet tundra), 8% (overgrown water), and 7% (water) in 2014 (Holl et al., 2019). Within the entire catchment (maximum estimated extent), these four surface classes amounted to 63% (dry tundra), 16% (wet tundra), 9% (overgrown water), and 11% (open water; Muster et al., 2012). Therefore, the distribution of tundra surface classes within the footprint of the EC flux is similar to the distribution of tundra surface classes within the catchment and the observed EC fluxes can be considered representative of vertical fluxes for the entire catchment.

The question remains whether our study period between June 8 and September 8 covers all relevant flux contributions from the catchment. At our study site, no large methane bursts have been observed during the soil-refreezing period in autumn as described by (Mastepanov et al., 2013) for their arctic fen site in Greenland. For a dataset from 2003, Wille et al. (2008) shows that mean daily methane emissions go gradually down between September and November. However, some peaks of higher methane emissions occur during stormy days during the refreezing period (probably triggered by turbulence-induced pressure

pumping). However, these higher emissions during very windy conditions are only at maximum about three times higher than base line emissions, thus, much less than the methane flux peaks observed by Mastepanov et al. (2013). An article analyzing a long-term methane flux dataset from Samoylov Island, which includes data from several autumn refreezing periods and furthermore data from deep winter, is currently under revision (Rößger et al., 2022). This so far unpublished more extensive dataset also shows no large autumn methane bursts. However, the article estimates that about 14% of the annual methane budget of the polygonal tundra is emitted during the refreezing period. Accounting for this additional emission would likely increase the relevance of $CH_4$ fluxes in an annual NECB.

In addition, the importance of lateral C fluxes could become more relevant with a longer observation period. Especially at the beginning of the study period, we observe high water discharge rates and high DOC concentration. Most likely, we do not cover the complete melting season with our study period; we clearly see in the data of outgoing short wave radiation that the snowmelt started 14 May. Relevant lateral C fluxes could have occurred directly at the beginning of the melting period, as it has been observed in a palsa and a bog in Northern Sweden (Olefeldt and Roulet, 2012). However, one could also argue that the observed high lateral C fluxes at the beginning of the study period should not be included in the NECB. These high lateral C fluxes are likely linked to C-bearing river water which flooded the catchment before the observations started and drained through the catchments' outflows at the beginning of the observation period. In the course of the observation period, the origin of dissolved C in the observed lateral runoff might shift from allochthonous to autochthonous sources. Due to the unknown characteristics of this possible shift in sources for dissolved carbon, we included all available lateral C flux observations in the NECB estimation. This inclusion of lateral C fluxes that are likely not part of the catchments' NECB increases the relevance of lateral C fluxes in the NECB estimation. Because we potentially overestimated the impact of lateral C export on the NECB, our conclusion of a very limited role of dissolved carbon for appears to be an understatement - lateral C export likely plays an even smaller role.

In case we would include the lateral C fluxes between 14 May and 2 June and assume that the DOC flux at our site would show a similar pattern as the DOC flux in Olefeldt and Roulet, 2012 (74% of DOC flux during snowmelt), we would have a max. annual DOC flux of 0.21 g m$^{-2}$. From DIC flux we would only expect a low contribution during the snow melt due to likely high water discharge rates during the snowmelt and the negative correlation between DIC concentration and water discharge rate. Therefore, the inclusion of possible snowmelt-DOC flux and DIC flux would change the absolute numbers of these fluxes, however, likely not change our conclusion regarding the influence of DOC flux or DIC flux on the NECB.

Due to the multitude of flux components some simplifications were applied and the uncertainty of the NECB was not quantified to its full extent. Most uncertainties have been described in section 2.9 and have been accounted for, however, more uncertainties might also arise from missing observations or gap-filling approaches. This study e.g. discounts the contributions of particulate organic carbon (POC), since we only found small differences between filtered (average 6.01 mg L$^{-1}$) and unfiltered water samples (average 6.07 mg L$^{-1}$) with respect to total carbon content. Thus, we suggest that POC would contribute only very little to the lateral C flux and therefore to the NECB. In this study we also include a gap-filled time series of the DIC concentration in the estimation of the NECB (see section 2.4). We assessed an agreement between the observed data and the independent testing subset as $R^2_{adj} = 0.79$. Therefore, this approach could increase the random uncertainty during the gap-

filled period. However, the large potential bias of the catchment assessment dominated the uncertainty of $F_{DIC}$ and the random uncertainty of the DIC concentration played only a minor role. Overall, we assume that these additional uncertainties do not significantly change the results of the estimated NECB and therefore also not the conclusion of this study.

## 335  5  Conclusions

At the polygonal tundra site in the Arctic Lena River Delta, which we investigated for this study, the net ecosystem carbon balance was periodically dominated by laterally exported dissolved carbon. The relative impact of these water-borne carbon losses on the total net ecosystem carbon balance was particularily high in the early and late growing season. During the Lena River spring flood, the in absolute and relative terms largest amounts of dissolved organic carbon were exported. In the late

vegetation period, the relatively high impact of lateral C fluxes can largely be explained by low net ecosystem exchange rates of carbon dioxide due to generally diminished plant activity. During the seasons when soils are refreezing (October–November) or completely frozen (December–May), water discharge and consequently lateral C export ceases. Therefore, we conclude that lateral C export is even less important for the annual NECB than for the growing season NECB.

The contribution of lateral C fluxes to the cumulative NECB decreased on Samoylov Island over the growing season and was,

in contrast to temperate and boreal ecosystems, negligible compared to cumulative vertical growing season carbon fluxes. We therefore conclude that the NECB of a polygonal tundra landscape is sufficiently described when only vertical flux measurements are performed. Only studies which describe short-term tundra C balances should take lateral C export into account, particularily during or immediately following snow melt. Furthermore, in regions with rapid landscape degradation, lateral C fluxes could play a different, more relevant role in an ecosystem's carbon balance.

*Data availability.*  The dataset of this study is published at Pangaea (Beckebanze et al., 2022b).

*Author contributions.*  Benjamin R. K. Runkle, Christian Wille, and Lars Kutzbach designed the experiments, and Benjamin R. K. Runkle, Christian Wille, David Holl, and Lars Kutzbach carried out the fieldwork. Benjamin R. K. Runkle, Josefine Walz, Lutz Beckebanze, and Lars Kutzbach developed the idea for the analysis, and Christian Wille provided processed eddy covariance data. The formal analysis and data visualization were done by Lutz Beckebanze. David Holl and Lars Kutzbach supervised it. Lars Kutzbach, Julia Boike, and Torsten Sachs

provided resources for the instrumentation. Lutz Beckebanze prepared the manuscript with contributions from all co-authors.

*Competing interests.*  The authors declare that they have no conflict of interest.

*Disclaimer.* This study was funded by the Deutsche Forschungsgemeinschaft (DFG, German Research Foundation) under Germany's Excellence Strategy – EXC 2037 'CLICCS - Climate, Climatic Change, and Society' – Project Number: 390683824, contribution to the Center for Earth System Research and Sustainability (CEN) of Universität Hamburg". Longterm measurements of $CO_2$ and $CH_4$ fluxes were supported by the projects "CarboPerm" (grant no. 03G0836A) and "KoPf" (grant no. 03F0764A), both funded by the German Federal Ministry of Education and Research (BMBF). Torsten Sachs and Christian Wille were supported by the Helmholtz Association of German Research Centres through a Helmholtz Young Investigators Group grant to Torsten Sachs (grant no. VH-NG-821).

*Acknowledgements.* We would like to thank the members of the Russian–German field campaign *LENA 2014*, especially Sandra Petersen (Universität Hamburg) for sampling analysis and the crew of the Russian research station Samoylov for logistical as well as technical support. We are grateful to Tim Eckhardt and Leonardo de Aro Galera for valuable discussions about the data analysis and the manuscript and Sarah Wiesner for being a great support as a Ph.D. advisory panel chair (all at the Universität Hamburg). We also thank Jean-Louis Bonne (Université de Reims Champagne-Ardenne) for allowing us to use the picture of the polygonal tundra.

## Appendix A: Supplementary Material

### A1 DOC concentration from a spectrometer probe

We used a multiple stepwise regression model (MSR; following Draper and Smith (2014)) to estimate $C_{DOC}$ from a spectrometer probe in order to obtain a longer time series of $C_{DOC}$ compared to the time series from the TOC analyzer. We compared the $C_{DOC}$ analyzed in the TOC analyzer with the absorbance bands from a spectrometer probe to find suitable absorbance bands to describe the DOC concentration. The spectrometer probe measured the absorbance ($a_\lambda$) of the sampled water probe between the 200 and 740 nm wavelength ($\lambda$) in 2.5 nm steps. In this analysis, we focused on the commonly used absorbance values between 250 and 740 nm as well as Ratio 1 ($a_{465}/a_{665}$) and Ratio 2 ($a_{255}/a_{365}$). Absorbance values below 250 nm were neglected due to possible interference with inorganic substances, following Avagyan et al. (2014). For the application of the MSR model, we split the data set of $C_{DOC}$ from the TOC analyzer and the absorbance values from the spectrometer probe into training (75%) and validation sets (25%). Details on the application of the MSR can be found in Avagyan et al. (2014). We applied the MSR model in Matlab R2019b using the *stepwisefit* function.

We used the following wavelengths and ratios as predictors for the DOC concentration at the three outflows: 250 nm and ratio 2 at outflow $A_1$; 250 nm, 300 nm, and 722.5 nm at outflow $A_2$; and 250 nm, 690 nm, and 712.5 nm at outflow B. In figure A3, the validation set of $C_{DOC}$ from the TOC analyzer and the spectrometer probe are shown.

**Table A1.** Mean daily flux components for each week (first 15 rows) and cumulative flux components (last row) of the NECB ($CO_2$-C, $CH_4$-C, DIC, and DOC) during the measurement period in 2014. Uncertainties from systematic errors are shown with an en dash (–) and uncertainties from random errors are denoted by the plus-minus symbol ($\pm$).

| | NECB | vert $CO_2$-C Flux | vert $CH_4$-C Flux | DIC Flux | DOC Flux | Unit |
|---|---|---|---|---|---|---|
| 2–7 June | 9.6 – 12.3 ($\pm$ 21) | -7.0 $\pm$ 21.1 | 3.6 $\pm$ 0.3 | — | 13.0 – 15.8 | mg m$^{-2}$ d$^{-1}$ |
| 8–14 June | 134 – 136 ($\pm$ 23) | 117.3 $\pm$ 23.0 | 5.94 $\pm$ 0.24 | 7.74 – 9.37 | 2,67 – 3,23 | mg m$^{-2}$ d$^{-1}$ |
| 15–21 June | -134 – -132 ($\pm$ 22) | -148.5 $\pm$ 22.2 | 8.29 $\pm$ 0.43 | 5.43 – 6.57 | 1,31 – 1,58 | mg m$^{-2}$ d$^{-1}$ |
| 22–28 June | -551 – -550 ($\pm$ 18) | -565.9 $\pm$ 17.6 | 10.00 $\pm$ 0.40 | 4.30 – 5.20 | 0,95 – 1,15 | mg m$^{-2}$ d$^{-1}$ |
| 29 June–5 July | -478 – -478 ($\pm$ 18) | -488.6 $\pm$ 17.8 | 8.10 $\pm$ 0.40 | 1.71 – 2.07 | 0,33 – 0,40 | mg m$^{-2}$ d$^{-1}$ |
| 6–12 July | -685 – -684 ($\pm$ 22) | -700.8 $\pm$ 22.3 | 12.13 $\pm$ 0.20 | 3.04 – 3.68 | 0,33 – 0,40 | mg m$^{-2}$ d$^{-1}$ |
| 13–19 July | -406 – -405 ($\pm$ 23) | -419.1 $\pm$ 23.0 | 8.97 $\pm$ 0.55 | 3.77 – 4.56 | 0,46 – 0,56 | mg m$^{-2}$ d$^{-1}$ |
| 20–26 July | -701 – -700 ($\pm$ 22) | -715.1 $\pm$ 21.7 | 7.90 $\pm$ 0.55 | 5.26 – 6.36 | 0,68 – 0,82 | mg m$^{-2}$ d$^{-1}$ |
| 27 July – 2 August | -220 – -219($\pm$ 27) | -237.1 $\pm$ 27.1 | 14.96 $\pm$ 0.65 | 2.06 – 2.49 | 0,23 – 0,28 | mg m$^{-2}$ d$^{-1}$ |
| 3–9 August | -355 – -354 ($\pm$ 31) | -370.5 $\pm$ 31.3 | 14.39 $\pm$ 0.23 | 1.22 – 1.47 | 0,10 – 0,13 | mg m$^{-2}$ d$^{-1}$ |
| 10–16 August | -23 – -22 ($\pm$ 31) | -42.2 $\pm$ 30.7 | 18.05 $\pm$ 0.43 | 1.36 – 1.64 | 0,12 – 0,15 | mg m$^{-2}$ d$^{-1}$ |
| 17–23 August | 16 – 17 ($\pm$ 27) | -1.0 $\pm$ 26.6 | 14.10 $\pm$ 0.36 | 2.84 – 3.44 | 0,42 – 0,51 | mg m$^{-2}$ d$^{-1}$ |
| 24–30 August | 253 $\pm$ 255 ($\pm$ 29) | 236.7 $\pm$ 29.1 | 13.68 $\pm$ 0.20 | 2.85 – 3.44 | 0,61 – 0,74 | mg m$^{-2}$ d$^{-1}$ |
| 31 August–6 September | 523 – 524 ($\pm$ 30) | 508.6 $\pm$ 30.2 | 11.27 $\pm$ 0.55 | 2.68 – 3.24 | 0,75 – 0,91 | mg m$^{-2}$ d$^{-1}$ |
| 7–8 September | 276 – 277 ($\pm$ 42) | 266.4 $\pm$ 41.8 | 6.23 $\pm$ 0.46 | 2.67 – 3.23 | 0,80 – 0,96 | mg m$^{-2}$ d$^{-1}$ |
| **Season Sum** (8 June– 8 September) | -17.6 – -17.5 ($\pm$ 1.2) | -19.0 $\pm$ 1.2 | 1.0 $\pm$ 0.02 | 0.31 – 0.38 | 0.06 – 0.08 | **g m$^{-2}$** |

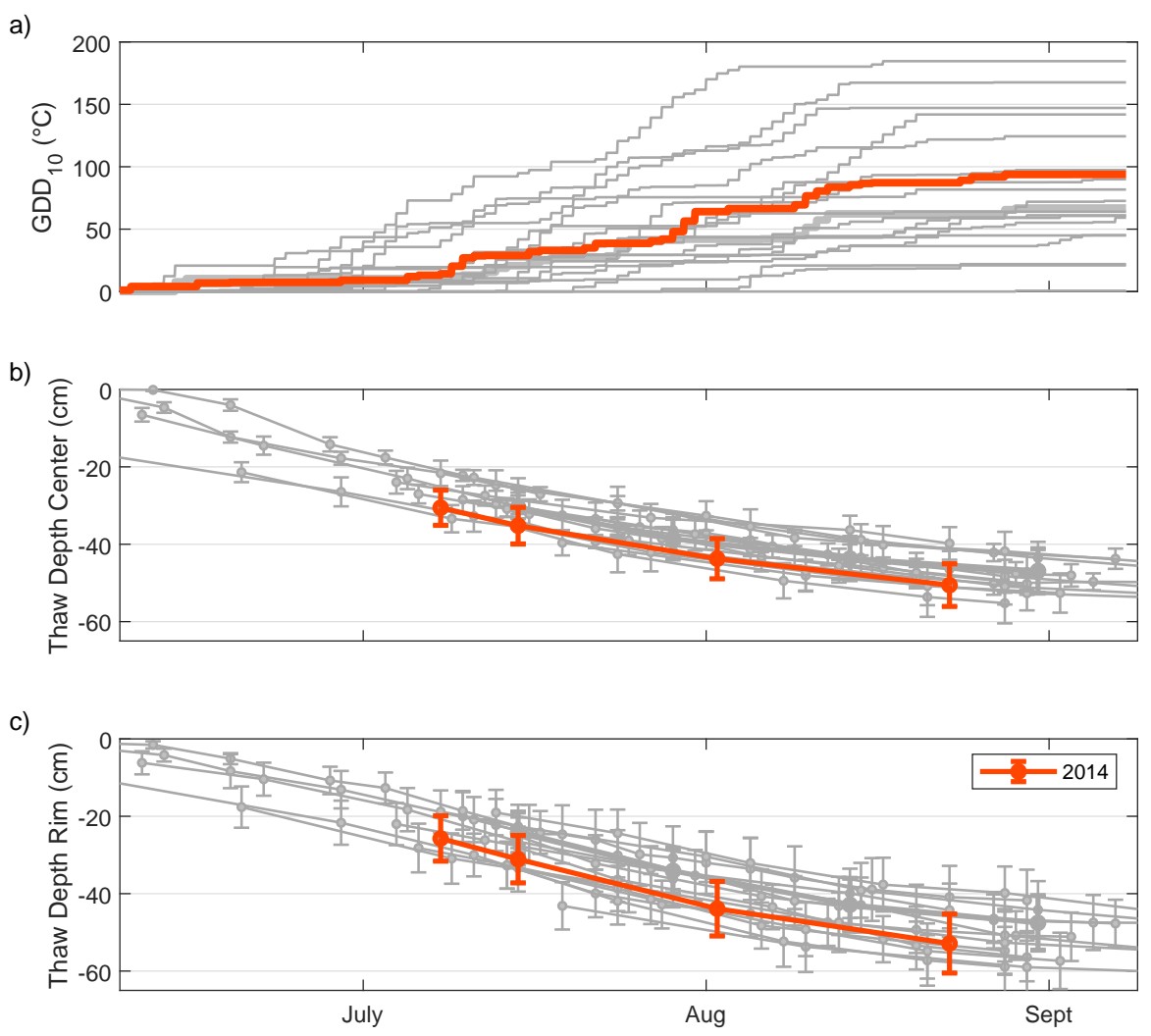

**Figure A1.** Cumulative growing degree days (GDD; (a) and thaw depths in the center and the rim of a polygon ((b) and (c), respectively) between 8 June and 8 September of 2002–2018. The year 2014 is highlighted in orange. All other years are displayed in gray.

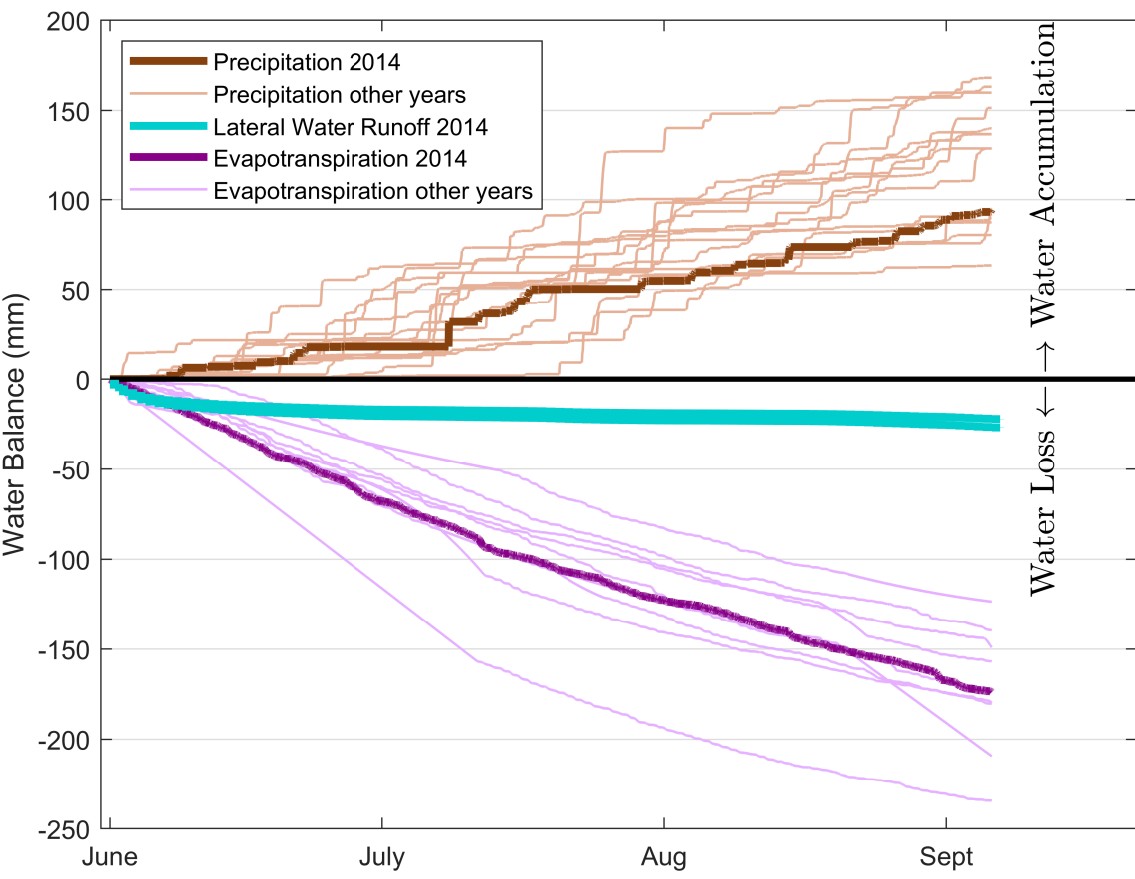

**Figure A2.** Water balance at the study site between 2 June and 8 September 2014 with cumulative precipitation amounts above zero and cumulative evapotranspiration and water runoff rate below zero. Cumulative precipitation and evapotranspiration for the years 1998-–2018 and 2007-–2018, respectively, are shown in lighter colors (with the exception of 2014, which is shown in darker colors).

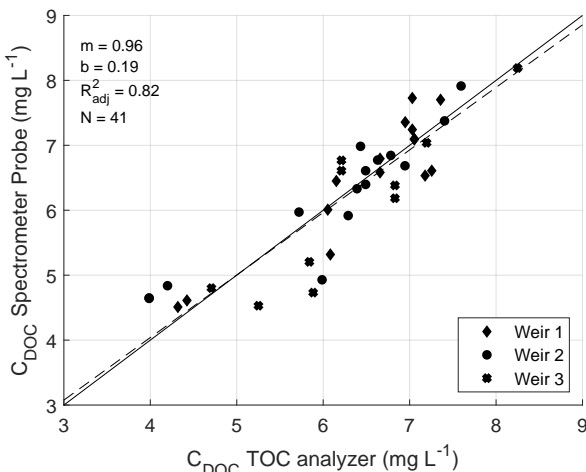

**Figure A3.** Validation set of $C_{DOC}$ from the TOC analyzer and the spectrometer probe with the corresponding linear regression (dashed line). The solid line represents the 1:1 line.

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
