# Peer review of "Lateral carbon export has low impact on the net ecosystem carbon balance of a polygonal tundra catchment"

_Biogeosciences, 2022_

## Author Comment (AC1)

**Reply to Referee 1**

Review on the manuscript titled 'Lateral carbon export has low impact on the net ecosystem carbon balance of a polygonal tundra catchment', submitted for publication in Biogeosciences by Beckebanze et al. in May 2022.

The presented study evaluates the contribution of lateral carbon exchange to the net ecosystem carbon balance of a small Arctic polygonal tundra catchment underlain by permafrost. Monitoring vertical and lateral flux rates over a period of about 3 months in summer 2014 reveals the temporal development of fCO2, fCH4, DOC and DIC, and how the relative contribution of each fraction changes over the course of the growing season. These datasets demonstrate that the lateral export pathways only make up about 2% of the vertical budget. Even though they may be important at the beginning of the season, or for shorter term studies, the authors thus conclude that lateral components can be neglected when targeting long-term carbon budgets.

The manuscript is well written and structured, and all conclusions are fully supported by the result material presented. The topic is certainly relevant, since as the authors correctly claim in their text lateral carbon export is considered to be a major component of the net carbon budget in certain Arctic ecosystems, but comprehensive studies that aim at assessing it are currently still lacking. Even though the presented material is a limited-scale case study, and extrapolation of the findings may be difficult, this is definitely a valuable contribution for this field of research. Besides some minor comments, which are listed further below, a have a few comments that should be addressed before the paper can be accepted for publication:

1.) You show in your results that the DOC flux is comparatively high in the first few days of June. This observation is also discussed at length in the discussion section. Now, from your description it sounded like the area was flooded by the Lena in the days before the start of the study period. This suggests that the standing water on the site, with high DOC, was previously laterally imported into the catchment. So if this carbon was not locally produced, in what way should it still be considered for the NECB? This should at least be considered in the discussion, and maybe you should tone down some of your statements regarding the elevated role of DOC early in the season.

> *Thank you for your kind words and thorough review.*
> *We expanded our discussion on the high DOC flux and stated, why we decided to include the high DOC flux from the beginning of the observation period into the NECB estimation.*

> Especially at the beginning of the study period, we observe high water discharge rates and high DOC concentration. Most likely, we do not cover the complete melting season with our study period. Relevant lateral C fluxes could have occurred directly at the beginning of the melting period, as it has been observed in a palsa and a bog in Northern Sweden (Olefeldt and Roulet, 2012). However, one could also argue that the observed high lateral C fluxes at the beginning of the study period should not be included in the NECB. These high lateral C fluxes are likely linked to C-bearing river water which flooded the catchment before the observations started and drained through the catchments' outflows at the beginning of the observation period. In the course of the observation period, the origin of dissolved C in the observed lateral runoff might shift from allochthonous to autochthonous sources. Due to the unknown characteristics of this possible shift in sources for dissolved carbon, we included all available lateral

C flux observations in the NECB estimation. This inclusion of lateral C fluxes that are likely not part of the catchments' NECB increases the relevance of lateral C fluxes in the NECB estimation. Because we potentially overestimated the impact of lateral C export on the NECB, our conclusion of a very limited role of dissolved carbon for appears to be an understatement - lateral C export likely plays an even smaller role.

2.) In section 4.1, you discuss the relationship between runoff, DOC/DIC, and precipitation. I think this discussion is not to the point. Since the source of DOC is obviously not the rain water, why should higher PRCP lead to higher DOC fluxes?

Your results suggest that, besides the initial period following the flooding in spring, DOC is leached from the thawing active layer, at a rate that is likely to be rather correlated to the increase in thaw depth than to PRCP. If this is the case, more PRCP would rather dilute the DOC, and thus influence the outgoing concentration, but not the exported DOC. Please adjust the discussion accordingly.

> *That is a valid point. We missed out on this part of the discussion and focused more on the DIC flux. Now we expanded the section on DOC concentration vs. water discharge rate in the discussion:*
>
> Similar to the DIC concentration, we also found a negative correlation between the water discharge rate and the DOC concentration when neglecting the period during the spring flood. This finding suggests that higher discharge rates dilute and decrease the DOC concentration. Therefore, in seasons with higher discharge rates, the DOC flux would not rise linearly and the contribution of DOC export to the NECB probably would not rise. A similar correlation between DOC concentration and the water discharge rate has been reported in a palsa a subarctic catchment (Olefeldt and Roulet, 2012).

Altogether, I consider my requested modifications as minor corrections. The manuscript is a valuable contribution to Biogeosciences, and I therefore recommend to accept it for publications after these small changes have been worked in.

> *Thank you!*

MINOR COMMENTS:

**l. 16f: I suggest to remove the last sentence from the abstract**

> *Thank you for this suggestion. We deleted the last sentence of the abstract.*

**l. 35f: I suggest to move the last sentence of paragraph 1 further down, to be the first sentence of paragraph 2.**

> *We followed your suggestion and moved the last sentence of paragraph 1 to the first sentence of paragraph 2.*

**l. 55: a better section header would be 'catchment characteristics'**

*We agree and changed accordingly.*

**l.77f: this sentence made me wonder how accurate a reference measurement with a bucket and a stop watch can actually be. So is this indeed helpful to correct the actual measurements? Maybe you can include this in the appendix where you treat the uncertainties of this part of the dataset in more detail.**

*Thank you for pointing this out. We used the wrong word in this context. The bucket measurement was used for validation of the observation. We changed the sentence accordingly.*

**l.99f: the implication of this gap-filling approach on your results should be treated as part of the discussion.**

*At the end of the discussion, we added a new paragraph on uncertainty estimation. In this section, we also discuss the gap-filling approach of the DIC concentration.*

Due to multiple flux components, the uncertainty of the NECB can not be quantified to the full extent. Most uncertainties have been described in section 2.10 and have been accounted for, however, more uncertainties might also arise from missing observations or gap-filling approaches. This study e.g. discounts the contributions of particulate organic carbon (POC), since we only found small differences between filtered (average 6.01 mg L$^{-1}$) and unfiltered water samples (average 6.07 mg L$^{-1}$) with respect to total carbon content. Thus, we suggest that POC would contribute only very little to the lateral C flux and therefore to the NECB. In this study we also include a gap-filled time series of the DIC concentration in the estimation of the NECB (see section 2.4). We assessed an agreement between the observed data and the independent testing subset as $R^2_{adj} = 0.79$. Therefore, this approach could increase the random uncertainty during the gap-filling period, however, the large potential bias of the catchment assessment dominated the uncertainty of F$_{DIC}$ and the random uncertainty of the DIC concentration played only a minor role. Overall, we assume that these additional uncertainties do not significantly change the results of the estimated NECB and therefore also not the conclusion of this study.

**Section 2.5: please provide information on how many samples were actually analyzed to produce the DOC results for this study**

*We added this information in the method section 2.5*

**Section 2.9: I found the description of the uncertainty treatment a bit vague in places. I therefore suggest to move some of the more detailed descriptions on uncertainty assessment from the appendix into the main text.**

*We agree with you. The description of the uncertainty estimation in section 2.9 was vague and not detailed. Therefore, we decided to delete most of the unspecific information in section 2.9 and move the whole uncertainty section from the appendix to the main text. We also split up the sections of cumulative fluxes and uncertainty estimation, so the uncertainty estimation moved to the new section 2.10.*

**L.154: you mention the 'vertical water balance', but include water runoff in the list of components ..??**

*Thank you for pointing this out. Of course, the runoff is not part of the vertical water balance, therefore we deleted the word "vertical" and also changed the figure A2 accordingly.*

**Figure 2: is water discharge the total discharge for the catchment, or the discharge normalized per unit area? If the latter, please adjust the units accordingly.**

*Thank you for this comment. The unit in Figure 2 was not correctly displayed. The unit should be L/s and we changed it accordingly.*

**Figure 4: even though the background pictures are nice generally, the chosen format makes the interpretation of the shown material rather difficult. I suggest to remove background figures altogether here (it can be left in Fig.3), and change this into a single panel figure with a line/bar chart showing the temporal development of the different flux components. Since you give the monthly values already in the table in the appendix, such a figure could be improved by a higher temporal resolution, e.g. weekly averages.**

*Thank you for this comment. We followed your suggestion and replaced this figure with a new bar-graph (see below). This graph shows mean daily fluxes with a weekly resolution and allows the reader to see the temporal development of the fluxes during the season. Additionally, we expanded the table in the appendix to weekly averages (the corresponding data to this graph).*

[Figure]

**ll.251ff: It would be good to mention also briefly in the methods section that you decided to neglect POC for this reason.**

*Thank you for this suggestion. We added in the section 2.9 the sentence:*

Other flux components of the lateral C flux, e.g. particulate organic carbon or particulate inorganic carbon, are not accounted for in this study.

**ll.254ff: this paragraph is kind of detached from the rest of the text, and particularly it is not connected to your own results.**

*We agree with you that this paragraph is detached. We therefore removed the paragraph. All information from this paragraph is already written in the introduction.*

---

## Author Comment (AC2)

**Reply to Referee 2**

This study by Beckebanze et al. seeks to improve estimates of soil organic carbon changes in permafrost-affected soils by estimating both vertical and lateral fluxes of carbon. They aim to achieve this with an extensive measurement campaign in a polygonal tundra ecosystem in Siberia, Russia. The study uses proven methods to estimate the carbon fluxes in a climatologically important area, where such measurements have not been done before. However, I find that the limited temporal coverage of the study (the growing season) puts into question the paper's main conclusions about the negligible importance of DIC and DOC export for the catchment NECB.

**General comments**

It is a bit puzzling to me why the carbon balance was computed for the growing season only. If the goal of the study is to improve estimates of soil carbon change in permafrost ecosystems, then an annual perspective including the highly dynamic spring and autumn seasons is required.

The majority of lateral transport will be in the snowmelt period in spring (65%-100% for dry tundra (palsa and bog habitats) in the study of Olefeldt et al., 2012). The authors state that the majority of the carbon export during the study period was accounted for by part of the spring flood in June, but that only part of the spring flood was covered by their measurements. This implies that the contribution of lateral DIC and DOC export to NECP may not be negligible on an annual basis, which appears to be one of the paper's main conclusions (see my comment about L10). Are there discharge measurements available for the spring season which would enable a rough estimation of the spring export of DOC and DIC? Or else, would it be possible to estimate spring discharge based on the annual water balance?

> *Thank you for your kind words and thorough review.*
> *Based on our limited data availability, we are not able to estimate the complete spring discharge rate. The observation only started on June 2. However, we clearly see in the data of outgoing short wave radiation that the snowmelt started May 14. In case we assume that the DOC flux at our site shows a similar pattern as the DOC flux in Olefeldt and Roulet, 2012 (74% of DOC flux during snowmelt), we would have a max. annual DOC flux of 0.21 g m$^{-2}$. From DIC flux we would only expect a low contribution during the snow melt due to likely high water discharge rates during the snowmelt and the negative correlation between DIC concentration and water discharge rate. Therefore, the inclusion of possible snowmelt-DOC flux and DIC flux would likely not change our conclusion regarding the influence of DOC flux or DIC flux on the NECB.*
> *In combination with the comments from second reviewer, we expanded the section on uncertainty of the measurements in the discussion and included the issue with water from the Lena river flowing out through the weirs. We also included that we do not know when exactly the shift from Lena water to ground- and melt water happened and therefore include observations from the beginning of the observation period:*

> Especially at the beginning of the study period, we observe high water discharge rates and high DOC concentration. Most likely, we do not cover the complete melting season with our study period. Relevant lateral C fluxes could have occurred directly at the beginning of the melting period, as it has been observed in a palsa and a bog in Northern Sweden (Olefeldt and Roulet, 2012). However, one could also argue that the

observed high lateral C fluxes at the beginning of the study period should not be included in the NECB. These high lateral C fluxes are likely linked to C-bearing river water which flooded the catchment before the observations started and drained through the catchments' outflows at the beginning of the observation period. In the course of the observation period, the origin of dissolved C in the observed lateral runoff might shift from allochthonous to autochthonous sources. Due to the unknown characteristics of this possible shift in sources for dissolved carbon, we included all available lateral C flux observations in the NECB estimation. This inclusion of lateral C fluxes that are likely not part of the catchments' NECB increases the relevance of lateral C fluxes in the NECB estimation. Because we potentially overestimated the impact of lateral C export on the NECB, our conclusion of a very limited role of dissolved carbon for appears to be an understatement - lateral C export likely plays an even smaller role.

Similar to the previous comment, large methane emissions may occur in the autumn as the gas is expunged from the freezing soil (Mastepanov et al., 2013). It would be interesting to know if such emissions occur on Samoylov Island and if so, they should probably be included in the carbon balance.

*Thank you for this comment. Yes, large methane emissions have been observed at this study site in the freezing period. We added a paragraph in the discussion on this topic and cited the observation data from this study site instead of Mastepanov et al. 2013:*

At our study site, no large methane bursts have been observed during the soil-refreezing period in autumn as described by Mastepanov et al. (2013) for their arctic fen site in Greenland. For a dataset from 2003, Wille et al. (2008) shows that mean daily methane emissions go gradually down between September and November. However, some peaks of higher methane emissions occur during stormy days during the refreezing period (probably triggered by turbulence-induced pressure pumping). However, these higher emissions during very windy conditions are only at maximum about three times higher than base line emissions, thus, much less than the methane flux peaks observed by Mastepanov et al. (2013). An article analyzing a long-term methane flux dataset from Samyolov Island, which includes data from several autumn refreezing periods and furthermore data from deep winter, is currently under revision (Rößger et al. 2022). This so far unpublished more extensive dataset also shows no large autumn methane bursts. However, the article estimates that about 14% of the annual methane budget of the polygonal tundra is emitted during the refreezing period. Accounting for this additional emission would likely increase the relevance of $CH_4$ fluxes in an annual NECB.

**Minor comments**

**L10: "annual fluxes": are annual totals of lateral fluxes compared to 93-day totals of vertical fluxes?**

*We deleted the word "annual".*

**L25: greenhouse gases -> greenhouse gases (GHGs)**

*We followed your suggestion and changed accodingly.*

**L34: NECB computations which include lateral transport are also available for the Stordalen Mire in subarctic Sweden (Lundin et al., 2016; Olefeldt and Roulet, 2012).**

*Thank you for this comment. We have missed out the NECB estimation of Lundin et al. 2016 and included it now into the introduction. However, we refrained from including Olefeldt and Roulet, 2012 in the introduction, since the NECB estimation is only mentioned in the discussion of the article without specific numbers.*

**L32: "basic Arctic landscape C balance models" what models are being referred to?**

*We were referring to landscape C models such as Koven et al. 2015 (CLM4.5BGC Model). However, this sentence has already caused some confusion in the internal review process, since this sentence was to unspecific, misleading, and not to the point of our research. Therefore, we decided to take this sentence out.*

**L135: "quantify the impact carbon losses due to lateral transport have on the total carbon balance of Samoylov Island": please describe briefly how the EC fluxes were extrapolated from the EC footprint to the entire island. I wonder whether it wouldn't make more sense to compare the catchment vertical fluxes to the catchment lateral fluxes.**

*Thank you for this suggestion. Our wording was misleading in this case, we do not observe the carbon balance of the whole island, but instead the area surrounding the EC tower. Therefore, we changed* "Samoylov Island" *to* "the chatchment" *and deleted the word* "total".

**L159: "The 2014 spring flood of the Lena River flooded parts of the catchment." Where is this information coming from?**

*This information is based on field observations by the co-author Benjamin Runkle. We added the source in the text.*

**L205: a detailed discussion of Lundin et al., 2016 and Olefeldt & Roulet, 2012 would be relevant here.**

*Thank you for this comment. We added a discussion on Olefeldt & Roulet, 2012 in the discussion and changed the discussion in this part according to a comment from reviewer 1. The new section is as followed:*

Similar to the DIC concentration, we also found a negative correlation between the water discharge rate and the DOC concentration when neglecting the period during the spring flood. This finding suggests that higher discharge rates dilute and decrease the DOC concentration. Therefore, in seasons with higher discharge rates, the DOC flux would not rise linearly and the DOC flux would affect the NECB only to a minor degree. A similar correlation between DOC concentration and the water discharge rate has been reported in a palsa a subarctic catchment (Olefeldt & Roulet, 2012).

**Figure 1d: Considering the discussion of the representativeness of the tower footprint, it would be helpful to be shown the average tower footprint (i.e. of the 2014 measurement period) on the map.**

*This is a valid point. We added the cumulative footprint representation as we did it in our previous paper (https://doi.org/10.5194/bg-19-1225-2022) and added the description of the different colors in the text of the figure:*

The cumulative footprint is shown in gray shades. 30% of the flux likely originated from within the dark gray area, 50% from within the medium-dark gray area, 70% from within the medium-light gray area and 90% from within the light gray area.

**Figure 5: this is a personal opinion, but I find the artificially coloured background images misleading. The water level and height of the vegetation would change over time.**

*Thank you for this comment. We replaced this figure with a new bar-graph (see below). This graph shows mean daily fluxes with a weekly resolution and allows the reader to see the temporal development of the fluxes during the season.*

[Figure]

**References**

Lundin, E. J., Klaminder, J., Giesler, R., Persson, A., Olefeldt, D., Heliasz, M., Christensen, T. R., and Karlsson, J.: Is the subarctic landscape still a carbon sink? Evidence from a detailed catchment balance, Geophys. Res. Lett., 43, 1988–1995, https://doi.org/10.1002/2015GL066970, 2016.

Mastepanov, M., Sigsgaard, C., Tagesson, T., Ström, L., Tamstorf, M. P., Lund, M., and Christensen, T. R.: Revisiting factors controlling methane emissions from high-Arctic tundra, 10, 5139–5158, https://doi.org/10.5194/bg-10-5139-2013, 2013.

Olefeldt, D. and Roulet, N. T.: Effects of permafrost and hydrology on the composition and transport of dissolved organic carbon in a subarctic peatland complex, J. Geophys. Res. Biogeosciences, 117, 1–15, https://doi.org/10.1029/2011JG001819, 2012.

---

## Author Response (AR1)

Dear Kees Jan van Groenigen,

Thank you very much for your reply and your positive feedback. We are happy to address your comment and added the text from the answer to reviewer 2 into the main text.

In line 304 we added the info that the snowmelt started on 14 May:

> Most likely, we do not cover the complete melting season with our study period; we clearly see in the data of outgoing short wave radiation that the snowmelt started 14 May.

And in line 317 we added the new paragraph on the relevance of DOC and DIC fluxes during the snowmelt period:

> In case we would include the lateral C fluxes between 14 May and 2 June and assume that the DOC flux at our site would show a similar pattern as the DOC flux in Olefeldt and Roulet, 2012 (74% of DOC flux during snowmelt), we would have a max. annual DOC flux of 0.21 g m$^{-2}$. From DIC flux we would only expect a low contribution during the snow melt due to likely high water discharge rates during the snowmelt and the negative correlation between DIC concentration and water discharge rate. Therefore, the inclusion of possible snowmelt-DOC flux and DIC flux would change the absolute numbers of these fluxes, however, likely not change our conclusion regarding the influence of DOC flux or DIC flux on the NECB.

We hope that this change in combination with the other previously changes in the review process are satisfactory.

Best regards,

Lutz Beckebaze